# Indications and endoscopic findings of upper gastrointestinal diseases in Africa: A systematic review & meta-analysis

Seid Mohammed Abdu *, Ebrahim Msaye Assefa, Hussen Abdu

Department of Biomedical Science, College of Medicine and Health Sciences, Wollo University, Dessie, Ethiopia

* seidmd041@gmail.com

## Abstract

### Background

Upper gastrointestinal endoscopy (UGIE) plays a crucial role in diagnosis of gastrointestinal pathology. Therefore, this systematic review and meta-analysis aimed to assess the indications and findings UGIE, while exploring their regional distribution and temporal trend across Africa.

### Methods

Systematic Reviews and Meta-Analysis of pooled prevalence for various indications and endoscopic findings were analyzed from multiple studies in accordance with Preferred Reporting Items for Systematic Reviews and Meta-Analysis (PRISMA) guidelines.

### Results

Seventeen common indication were identified. Of these dyspepsia was the most prevalent indication 52.4%, followed by abdominal pain 17.4%, hematemesis 13.9%, and GERD symptoms 11.2%. Other indications included dysphagia 9.2%, vomiting 9.2, odynophagia 3.5%, and melena 6.2% were identified. Rare indications such as anemia 2.3%, weight loss 2.6% were also reported. Regarding endoscopic findings, thirty-one common findings were identified by UGIE. Gastritis (33.3%) was the most common findings followed by normal findings 21.8%, the third most common was PUD 15.1%, particularly duodenal ulcer (10%), gastric cancer 3.3% were also prevalent in stomach. Related to esophageal findings, GERD 9.6%, esophagitis 8.3%, esophageal varices 7.2% and esophageal cancer 6.1% were identified. Regional difference were apparent, with esophageal cancer prevalent in Eastern (10%) and Southern Africa (10%). Gastritis (45%) and GERD (18%) were more apparent and common in Northern Africa. Even though it is not significant, temporal trends showed an increase in prevalence of gastritis (26 to 36%) and esophagitis (6 to 10%) from 2000-2010 to 2011-2024.

### Conclusion

Most UGIE indications resulted significant UGIT pathology. However, this analysis did not assess age, sex based indications and findings and their relationship among specific

**Data availability statement:** All relevant data are within the paper and its Supporting Information files.

**Funding:** The author(s) received no specific funding for this work.

**Competing interests:** The authors have declared that no competing interests exist.

**Abbreviation:** PUD, peptic ulcer disease; MeSH, medical subject heading; UGIE, upper gastro intestinal endoscopy; UGID, upper gastrointestinal disease; DU, duodenal ulcer; DU, gastric ulcer; GERD, gastroesophageal reflux disease.

indications and UGIE findings. So, future analysis should focus on age and sex based difference in indications and findings, and explore their relationship among specific indication and corresponding UGIE findings.

## Introduction

The gastrointestinal tract (GIT) runs from the mouth to the anus and is conventionally split into the upper mouth to the ileum and lower cecum to anus sections [1,2]. Embryologically, it is divided into the upper mouth to the duodenal papilla, middle duodenal papilla to the mid-transverse colon, and lower mid-transverse colon to anus, corresponding to the foregut, midgut, and hindgut [3–5].

Symptoms of GIT disorders rank among the most frequently reported complaints for primary care visits globally [6,7]. These conditions cause 105 million outpatient visits, 14 million hospital admissions, 236,000 deaths, and an annual cost of $142 billion in the US alone [8]. With over 17.7 million gastrointestinal endoscopic procedures carried out annually, GI symptoms make up 68% of all endoscopic procedures [9].

UGIE is indicated for symptoms such as persistent upper abdominal symptom despite an appropriate trial of therapy, and when it associated with anorexia and weight loss, or new-onset symptom in patients over 50 years. It is also warranted for active or recent upper GI bleeding with or without anemia, odynophagia, dysphagia, esophageal reflux symptoms that persist or recur despite appropriate therapy, persistent vomiting of unknown cause. Other indications including GI pathology that might affect management, such as prior ulcer, or GI bleeding before organ transplantation or long-term medication use. Additionally, UGIE is indicated for familial adenomatous polyposis, radiologically suspected neoplastic lesions, ulcers or stricture. It is also used for tissue of fluid sampling, in selected patients with suspected portal hypertension to assess or treat esophageal varices, to evaluate acute injury from caustic ingestion [10]. When alarm symptoms are generally accepted, there should be no hesitation in performing an endoscopy [10–12]. UGIE also called esophagogastroduodenoscopy (EGD) is a useful diagnostic and therapeutic tool for conditions affecting the esophagus, stomach, and upper portions of the duodenum [13,14]. Endoscopies are not only useful for direct inspection; they can also be used for certain therapeutic interventions like banding, sclerotherapy, polypectomy, stricture stretching, and biopsies from suspicious lesions [15,16]. The major conditions identified by UGIE include gastroesophageal reflux disease (GERD), esophageal varicose, PUD, and upper gastrointestinal cancers [14]. However, to the best of our knowledge, there has been no systematic assessment of endoscopic indications and findings of upper GI disease in Africa. Therefore, this systematic review and meta-analysis aims to provide a comprehensive understanding of UGIE indications and findings in Africa by analyzing their pooled prevalence, explores regional variation across the continent, trend over time and compares the burden and patterns of these indications and findings with global data.

## Methods

### Protocol registration

The purpose of this systematic review and meta-analysis was to compile the body of knowledge regarding the indications and endoscopic findings of UGID in Africa. Under the registration number (CRD42024554218), the protocol has been registered with the International Prospective Register of Systematic Reviews (PROSPERO).

## Search strategy

The Preferred Reporting Items for Systematic Reviews and Meta-Analyses (PRISMA) guidelines were followed in the conduct of this systematic review and meta-analysis [17] (S1 Checklist). Up until December 31, 2024, a thorough literature search was conducted using all available electronic databases, including PubMed, Google Scholar, and Hinari. Three steps were involved in the search strategy. The first step involved finding pertinent Medical Subject Headings (MeSH) and other terms in the literature. Complete searches were carried out in the aforementioned databases during the second phase. In the third stage, university websites and the bibliographies of pertinent studies were examined for the existence of studies that qualified. Various MeSH terms and keywords were combined using a Boolean operator to search the databases (OR, AND) (S1Appendix).

## Criteria for considering studies for the review

**Inclusion criteria.** We included observational studies conducted in African countries that reported on indications for or findings of endoscopy within the study population. Additionally, only articles on English and published between January 1, 2000, and December 31, 2024, were considered eligible. Studies were required to use endoscopy as diagnostic method for evaluating UGID. To ensure broader applicability, minimizing selection bias, and enhance comparability of findings, we excluded studies that focused on patients with specific indications, such as upper gastrointestinal bleeding or dyspepsia alone, as well as studies limited to particular age groups, including children, adolescent, or older adults.

**Exclusion criteria.** Case reports, case series studies, and any other research lacking the necessary data to report indications or endoscopic findings of UGID are not included in the selection process.

## Categorization and terminologies

We categorized dyspepsia in accordance with Rome IV criteria, which includes symptoms such as epigastric pain, early satiety, and postprandial fullness. In line with Rome IV criteria, epigastric pain was considered as part of dyspepsia, and we treated them as a combined indication for UGIE. Additionally, heart burn and regurgitation, both are symptoms of GERD, were classified under GERD symptoms as indication for endoscopy. During the data extraction process, we carefully reviewed each study to verify that symptoms such as epigastric pain, dyspepsia, early satiety, and postprandial fullness were accurately reported and did not overlap within individual subjects. After careful checking for overlap in the symptoms, thereby minimizing potential bias and avoiding overestimation bias, we combined these indicators into the broader category of dyspepsia and GERD symptoms in Rome IV criteria [18,19].

## Data extraction and quality assessment

The articles' eligibility was evaluated independently by three investigators (SMA, EMA, HA). The same three writers used Microsoft Excel to separately extract data. Author name, publication year, country, UGID symptoms and indication, sample size, number of patients diagnosed by endoscopy, and endoscopic evaluation results were all included in the data extraction sheet. Disagreements among investigators were settled through deep re-evaluation of articles and decision made agreement. Titles and abstracts of the identified articles were examined to find studies on the indications and endoscopic outcomes. Articles that the title and abstract deemed relevant were vetted for complete eligibility. Based on the endoscopic results and indications of the studies, the methodological quality of the included studies was evaluated using the prevalence JBI quality assessment tool (S2 Appendix) [20].

## Outcome of interest

The main goal of this systematic review and meta-analysis was to determine the most common indications, endoscopic findings of UGID, regional variation, and temporal trend which were originally reported in the paper as a percentage or as the number of cases (n) out of all the patients who were assessed (N).

## Statistical analysis

Through the use of the random-effects inverse variance method, the pooled prevalence of indications and endoscopic findings of UGID in Africa was determined, along with a corresponding 95% CI. Assessments of heterogeneity were also conducted. Cochran's Q test and $I^2$ test statistics were used to assess the heterogeneity of the studies. STATA version 17 (STATA Corporation, College Station, TX, USA) was used for the statistical analyses.

## Ethical approval and consent to participate

Not applicable since the datasets used and/or analyzed during the current study are freely available on the database and website

# Results

In this systematic review and meta-analysis, 68 studies were included to pool the indications and findings of UGIE related to UGID (Table 1). A search of electronic databases including PubMed, Hinari, and Google Scholar retrieved 2136 articles; with an additional 14 records from search engine (Yahoo, google). After removing 239 duplicates and 773 records for other reasons, 1124 records from databases were screened, resulting to the exclusion of 547 irrelevant papers. Subsequently, 577 reports were sought for retrieval, of which 251 from databases were assessed for eligibility. Following exclusion various reason, this systematic review and meta-analysis included 66 from databases and 2 studies from other sources, in total, 68 studies were included in the review, corresponding to 34 reports of included studies (S1 Fig).

## Characteristic of studies

A total of 68 institution-based observational studies with a population of 120460 were included in the meta-analysis. Of the 68 studies, 25 were carried out in East Africa, and 22 were recovered from West Africa; while 11 studies were from North Africa, and 9 South Africa. Only 1 study was retrieved from Central Africa. Eight studies were conducted in Ghana, elven studies were carried out in Nigeria, six in Ethiopia, and ten in Egypt. The remaining studies were conducted in their respective countries. Of the included studies, 40 (60.42%) used retrospective cross-sectional studies as the main study design, and the remaining 28 (58.82%) used prospective cross-sectional studies. Endoscopy was used as the only screening method in the included studies (Table 1).

## Risk of bias assessment

The synthesis is highly reliable, with 79.4% of studies at low risk of bias, ensuring robust methodological quality. Moderate-risk studies (16.2%) introduce minor variability due to issues like small sample, while high-risk (4.4%) had negligible impact due to limited number and weight (Table 1).

**Table 1. Characteristics of the included studies on indication and finding of UGIE.**

| First Author name, Year of publication | Countries | Region | Study design | Years of span | Sample size | Male | Female | Risk of bias |
|---|---|---|---|---|---|---|---|---|
| Argaw, A.M., et al. 2023 [21] | Ethiopia | East Africa | Retrospective | 2012-2019 | 5753 | 3648 | 2105 | Low |
| Assefa, B., et al. 2022 [22] | Ethiopia | East Africa | Prospective | 2020 | 218 | 118 | 100 | Low |
| Melak W, et al. 2023 [23] | Ethiopia | East Africa | Prospective | 2018-2022 | 142 | 75 | 67 | Moderate |
| Kiros YK et al.2017 [24] | Ethiopia | East Africa | Retrospective | 2011-2015 | 1994 | 1170 | 824 | Low |
| Getahun GM et al. (2015 [25] | Ethiopia | East Africa | Retrospective | 2005-2015 | 1310 | 668 | 642 | Low |
| Zena D et al. 2024 [26] | Ethiopia | East Africa | Retrospective | 2023-2024 | 279 | 118 | 161 | Low |
| Makanga W, et al. 2014 [27] | Kenya | East Africa | Retrospective | 2011-2013 | 5948 | 1372 | 1564 | Low |
| Mwangi CN. et al. 2020 [28] | Kenya | East Africa | Prospective | 2018-2019 | 487 | 266 | 221 | Low |
| Ayuo PO, et al. 2014 [29] | Kenya | East Africa | Retrospective | 1993-2003 | 1690 | 864 | 826 | Low |
| Lodenyo H. et al. 2005[ 30] | Kenya | East Africa | Retrospective | 1998-2001 | 768 | 484 | 284 | Low |
| Adani AA et al. 2023[31] | Somalia | East Africa | Retrospective | 2021-2022 | 634 | 363 | 271 | Low |
| Bulur O et al. 2018 [32] | Somalia | East Africa | Retrospective | 2015-2017 | 306 | 209 | 97 | Low |
| Obayo S. et al. 2015 [33] | Uganda | East Africa | Prospective | 2014-2015 | 184 | 110 | 74 | Moderate |
| Namugerwa J. et al. 2017 [34] | Uganda | East Africa | Retrospective | 2017 | 385 | 151 | 234 | Moderate |
| Okello TR, et al. (2016) [35] | Uganda | East Africa | Retrospective | 2015 | 605 | 243 | 362 | Low |
| Abeshouse MA, et al. 2024 [36] | Uganda | East Africa | Retrospective | 2020-2022 | 333 | ------ | ------ | Moderate |
| Doe MJ et al.(2021) [37] | Uganda | East Africa | Retrospective | 2009-2019 | 833 | 474 | 359 | Low |
| Walker TD et al 2014 [38] | Rwanda | East Africa | Retrospective | 2011-2014 | 961 | 438 | 523 | Low |
| Ayana SM et al. 2014 [39] | Tanzania | East Africa | Prospective | 2009-2010 | 208 | 99 | 109 | Low |
| Qu LS, et al. 2023 [40] | Tanzania | East Africa | Retrospective | 2013-2021 | 3146 | 1455 | 1691 | Low |
| Khamisi R H. 2013 [41] | Tanzania | East Africa | Prospective | 2013 | 159 | 94 | 65 | Low |
| Said EM et al. 2014 [42] | Sudan | East Africa | Prospective | 2013 | 30 | 19 | 11 | High |
| El Shallaly et al.2021 [43] | Sudan | East Africa | Prospective | 2007-2019 | 1859 | 1058 | 794 | Low |
| Elhadi AA et al. 2014 [44] | Sudan | East Africa | Prospective | 2013 | 390 | 170 | 220 | Low |
| Adam HY et al. 2008 [45] | Sudan | East Africa | Retrospective | 2003-2007 | 1150 | 656 | 494 | Low |
| Yahya H. 2023 [46] | Nigeria | West Africa | Retrospective | 2014-2022 | 1958 | 1339 | 619 | Low |
| Ray-Offor E. et al. 2020 [47] | Nigeria | West Africa | Prospective | 2014-2019 | 434 | ------ | ------ | Low |
| Okoye OG. et al.2021 [48] | Nigeria | West Africa | Prospective | 2016-2017 | 132 | 66 | 66 | Moderate |
| Odeghe E A. et al. 2023 [49] | Nigeria | West Africa | Retrospective | 2020-2021 | 227 | 96 | 131 | Low |
| Obonna GC et al. 2020 [50] | Nigeria | West Africa | Retrospective | 2012-2020 | 264 | ------ | ------ | Moderate |
| Ismaila BO. et al. 2013 [51] | Nigeria | West Africa | Prospective | 2010-2012 | 122 | ------ | ------ | Moderate |
| Misauno M. et al. 2011 [52] | Nigeria | West Africa | Retrospective | 1999-2010 | 989 | 593 | 396 | Low |
| Ngim O et al. 2017 [53] | Nigeria | West Africa | Prospective | 2012-2014 | 171 | 86 | 85 | Low |
| Jeje EA et al. 2013 [54] | Nigeria | West Africa | Prospective | 1994-1997 | 184 | 101 | 83 | Low |
| Nwokediuko SC et al. 2012 [55] | Nigeria | West Africa | Retrospective | 1995-99, 2006-10 | 1365 | 727 | 638 | Low |
| Oluwagbenga OO et al [56] | Nigeria | West Africa | Retrospective | 2003-2007 | 181 | 95 | 86 | Moderate |
| Archampong TN. et al.2016 [57] | Ghana | West Africa | Prospective | 2010-2012 | 242 | 127 | 115 | Low |
| Darko R et al 2015 [58] | Ghana | West Africa | Retrospective | 1999-2012 | 2401 | 1120 | 1281 | Low |
| Agyei-NkansahA et al. 2019 [59] | Ghana | West Africa | Prospective | 2012 | 371 | 159 | 212 | Low |
| Duah A et al.2022 [60] | Ghana | West Africa | Retrospective | 2019-2020 | 571 | 244 | 327 | Low |
| Aduful HK. et al. 2007 [61] | Ghana | West Africa | Retrospective | 1995- 2002 | 6977 | 3777 | 3200 | Low |
| Gyedu A, and Yorke J 2014 [62] | Ghana | West Africa | Retrospective | 2006-2011 | 3110 | 1327 | 1783 | Low |
| Dakubo JC et al. 2011 [63] | Ghana | West Africa | Prospective | 2008 | 1643 | 792 | 851 | Low |
| Tabiri S et al.2015 [64] | Ghana | West Africa | Retrospective | 2010-2014 | 2414 | 1199 | 1215 | Low |
| Koura M. et al.2017 [65] | Burkina Faso | West Africa | Prospective | 2015-2016 | 1022 | 470 | 552 | Low |

*(Continued)*

**Table 1.** (Continued)

| First Author name, Year of publication | Countries | Region | Study design | Years of span | Sample size | Male | Female | Risk of bias |
|---|---|---|---|---|---|---|---|---|
| Meda ZC et al. 2023 [66] | Burkina Faso | West Africa | Prospective | 2019-2020 | 180 | 96 | 84 | Low |
| Okon JB et al. 2021 [67] | Ivory Cost | West Africa | Prospective | 2019-2020 | **1010** | 475 | 535 | Low |
| Gado A. et al. 2015 [68] | Egypt | North Africa | prospective | 2000-2013 | 4477 | ------ | ------ | Low |
| El-Ghannam R et al. 2019 [69] | Egypt | North Africa | Prospective | 2019 | 95 | 37 | 58 | High |
| Gomaa AA et al. 2022 [14] | Egypt | North Africa | Retrospective | 2018-2020 | 2281 | 1138 | 1143 | Low |
| Elbadry M et al. 2024 [70] | Egypt | North Africa | Retrospective | 2016-2021 | 4433 | 2570 | 1863 | Low |
| Abdelrazek FG et al. 2024 [71] | Egypt | North Africa | Prospective | 2024 | 400 | 224 | 176 | Low |
| Raafat KM. et al 2022 [72] | Egypt | North Africa | Prospective | 2022 | 100 | 50 | 50 | Moderate |
| Yasser MY et al. 2023 [73] | Egypt | North Africa | Prospective | 2021-2022 | 125 | +79 | 142 | Low |
| Moustafa HM, et al. 2023 [74] | Egypt | North Africa | Retrospective | 2019-2020 | 2500 | 1226 | 1274 | Low |
| Fouad M et al. 2018 [75] | Egypt | North Africa | Retrospective | 2013-2015 | 218 | 128 | 90 | Low |
| Ali MH et al. 2024 [76] | Egypt | North Africa | Retrospective | 2018-2019 | 928 | 536 | 392 | Low |
| Tumi A. et al.2007 [77] | Libya | North Africa | Prospective | 2000 | 99 | 53 | 46 | High |
| Cheddie S.et al.2020 [78] | South Africa | South Africa | Retrospective | 2014-2016 | 1000 | 306 | 694 | Low |
| Mnyombolo Y et al. 2022 [79] | South Africa | South Africa | Retrospective | 2017-2018 | 300 | ------ | ------ | Low |
| Ntola VC et al. 2019 [80] | South Africa | South Africa | Retrospective | 2015 | 194 | 73 | 121 | Moderate |
| Fernando N et al. 2001 [81] | Zambia | South Africa | Prospective | 1999-2002 | 191 | ------ | ------ | Moderate |
| Kayamba V, et al. [82] | Zambia | South Africa | Retrospective | 1977-2021 | 25849 | ------ | ------ | Low |
| Kelly P et al.2008 [83] | Zambia | South Africa | Retrospective | 1999-2005 | 2132 | 1100 | 941 | Low |
| Kayamba V. et al 2015 [84] | Zambia | South Africa | Retrospective | 1977-2015 | 16,953 | 8820 | 6593 | Low |
| Wolf LL. et al. 2012 [85] | Malawi | South Africa | Prospective | 2008-2010 | 1004 | 562 | 441 | Low |
| Mothes H. et al. 2009 [86] | Malawi | South Africa | Retrospective | 2004-2006 | 441 | ------ | ------ | Low |
| Adonis NM et al.2021 [87] | Congo(DRC) | Central Africa | Retrospective | 2014-2016 | 1000 | 450 | 550 | Low |

## Indications for upper gastrointestinal endoscopy

Based on this systematic review and meta-analysis of 68 studies, 17 common indication for UGIE were identified. Out of 17 indications, dyspepsia was the most frequently reported indications, retrieved from 55 studies, with a pooled prevalence of 52.4% (44.9, 61). Hematemesis was the other common indication, reported in 47 studies, with a pooled prevalence of 13.9% (11.9, 15.9) (S2 Fig). GERD symptoms were the other most common, reported in 34 studies, with a pooled prevalence of 11.2% (9.6, 12.9) (S3 Fig). Dysphagia was another significant indications, reported in 40 studies, with a pooled prevalence of 9.2% (8.4, 10.5) (S4 Fig). Vomiting was reported in 40 studies as well, with a pooled prevalence of 9.7% (8.2, 11.2). Abdominal pain, with a pooled prevalence of 17.4% (13, 21.8), was reported in 17 studies, making it one of the most common reason of UGIE. Less common indication included anemia (22 studies, 6.2% (3.9, 8.4), melena (17 studies, 6.2 (3.9, 8.4), and odynophagia (12 studies, 3.5% (1.8, 5.2). Rare indication were weight loss (15 studies, 2.6% (1.8, 3.3), ascites (5 studies, 3.1% (1.2, 4.9) and bloating (4 studies 6.2% (0.4, 12.1). Other miscellaneous indications were reported by 41 studies, with pooled prevalence of 10.8% (9, 12.6). The I² values for the pooled estimates showed substantial heterogeneity across studies, ranging from 69.8% to 99.9%, with the majority of indications exceeding 90%, indicating high variability in the reported prevalence of UGIE indications among studies (Table 2).

**Endoscopic findings on upper gastrointestinal tracts.** A total of 31 most common findings were identified by UGIE. The most frequent finding in this review was gastritis,

**Table 2. The pooled prevalence of different indication of UGIE patients in Africa 2024.**

| Indications | Number of studies | Number of participants | Prevalence (95% CI) | I² | p-value |
|---|---|---|---|---|---|
| Dysphagia | 40 | 63177 | 9.2 (8, 10.5) | 99% | <0.001 |
| GERD* symptoms | 34 | 46721 | 11.2(9.6, 12.9) | 98.9% | <0.001 |
| Odynophagia | 12 | 8981 | 3.5(1.8, 5.2) | 96.6% | <0.001 |
| Dyspepsia | 55 | 80927 | 52.4(44.9, 61) | 99.9% | <0.001 |
| Ascites | 5 | 1650 | 3.1 (1.2, 4.9) | 69.8% | <0.001 |
| Hematemesis | 47 | 78485 | 13.9 (11.9, 15.9) | 99.1% | <0.001 |
| Weight loss | 15 | 24556 | 2.6 (1.8, 3.3) | 93.5% | <0.001 |
| Chest pain | 3 | 3186 | 5(-1.9,11.9) | 97.3% | <0.001 |
| Abdominal pain | 17 | 29922 | 17.4(13, 21.8) | 99.6% | <0.001 |
| Anemia | 22 | 49945 | 2.3(1.7, 2.9) | 95.4% | <0.001 |
| Melena | 17 | 26084 | 6.2 (3.9, 8.4) | 99% | <0.001 |
| Vomiting | 40 | 48026 | 9.7 (8.2. 11.2) | 98.7% | <0.001 |
| Anorexia | 5 | 3983 | 9.2(3.4, 15) | 97.9% | <0.001 |
| Bloating | 4 | 961 | 6.2 (0.4, 12.1) | 93.9% | <0.001 |
| Belching | 2 | 2888 | 1.1(0.7, 1.5) | 1.1% | 0.315 |
| Suspicion of Cancer in stomach | 9 | 9950 | 4.4(2.3, 5.2) | 96.4% | <0.001 |
| Suspicion of Cancer in esophagus | 5 | 6910 | 4.1(1.8, 6.1) | 97.6% | <0.001 |
| Other | 41 | 42879 | 10.8(9, 12.6) | 99.4% | <0.001 |

reported in 33.3% of cases based on 53 studies. PUD was also the most prevalent findings, appearing in 59 studies with a pooled prevalence of 15.1%. This was followed by normal endoscopic finings, documented in 55 studies and observed in 21.8% of case. There were significant heterogeneities observed for PUD, normal endoscopic finings, and gastritis, with I² values of 98.7%, 99.7% and 99.6% respectively.

In the esophagus, the most common endoscopic findings was GERD, identified in 9.6% of cases across 31 studies (S5 Fig). The second and third most frequent esophageal findings were esophagitis and esophageal varices, reported in 8.3% & 7.2% of case from 40 & 42 studies respectively. Esophageal cancer was the fourth most common, with prevalence of 6.1% reported in 35 studies (S6 Fig). In stomach, gastric cancer was the other frequent gastric findings, documented in 3.3% across 43 studies. In the duodenum, DU was the most frequent findings, reported 46 studies with pooled prevalence of 10%, duodenitis followed, documented in 39 studies with prevalence of 10.9% (Table 3).

## Regional distribution and trends of UGIE findings and indications

The review revealed that some endoscopic findings and their indications exhibited regional and overtime variation across Africa, while others remained consistent. Beginning with esophageal findings, esophageal cancer was the most prevalent in Eastern Africa and Southern Africa, with pooled prevalence of 10% (8–13) and 10% (8–12), respectively, while Wester Africa only 1% (0–1), and no data were available for Northern Africa. Esophagitis was highest in Northern Africa (19%(11–27) compared to Eastern (9% (7–11) and Western Africa (8% (5–10). Similarly, esophageal varices was most frequent in North Africa (18% (7-28), with lower rate observed in Western 5%, Eastern 8%, and Southern Africa 6%. Gastritis emerged as the most common endoscopic findings across all regions, with the prevalence of in Northern Africa 45%, followed by Western 33%, and Eastern 29%. PUD showed variable prevalence, ranging from 8% in Southern Africa to 19% of Western Africa. Regarding, gastric cancer,

**Table 3. The pooled prevalence of different UGIE findings in Africa 2024.**

| Endoscopic findings | Number of studies | Number of participants | Pooled Prevalence (95% CI) | I² | p-value |
|---|---|---|---|---|---|
| PUD* | 59 | 108017 | 15.1 (13.2, 16.9) | 98.7% | <0.001 |
| GU** | 48 | 101436 | 5.7 (4.7,6.7) | 98.4% | <0.001 |
| DU*** | 46 | 101189 | 10 (8.6, 11.3) | 98.2% | <0.001 |
| Both GU &DU | 8 | 36887 | 0.6 (0.2, 0.9) | 92.4% | <0.001 |
| Gastritis | 53 | 57906 | 33.3 (28.5, 38) | 99.6% | <0.001 |
| Gastric polyp | 21 | 17526 | 0.7 (0.5, 0.9) | 40.5% | 0.029 |
| Gastric mass | 7 | 5448 | 2.4 (1.2, 3.7) | 90% | <0.001 |
| Gastric atrophy | 8 | 10178 | 15.5 (10.6, 20.5) | 99.5% | <0.001 |
| Gastric erosion | 13 | 25130 | 8 (6.2, 9.8) | 99.1% | <0.001 |
| GOO**** | 17 | 45511 | 3.8 (3, 4.7) | 96.2% | <0.001 |
| Gastric tumor | 8 | 17921 | 3.1 (2.2, 3.9) | 88.4% | <0.001 |
| Gastric cancer | 43 | 84759 | 3.3 (2.8, 3.8) | 92.7% | <0.001 |
| Portal hypertensive gastropathy | 9 | 15338 | 8 (4, 12) | 99.5% | <0.001 |
| Foreign body | 8 | 11359 | 0.5 (0.2, 0.9) | 78.2% | <0.001 |
| Duodenitis | 39 | 48674 | 10.9(8.7, 13) | 99.3% | <0.001 |
| Pyloric obstruction | 9 | 11793 | 1.3(0.9, 1.7) | 50.7% | 0.039 |
| Normal endoscopic findings | 55 | 89399 | 21.8 (17.5, 26.1) | 99.7% | <0.001 |
| Gastro-duodenitis | 6 | 7325 | 12.4 (1.7, 23.1) | 99.7% | <0.001 |
| Duodenal cancer | 6 | 14591 | 0.2 (0, 0.3) | 54.7% | 0.051 |
| Deformed duodenal bulb | 4 | 8190 | 1.9 (0.6, 3.3) | 92.6% | <0.001 |
| Esophageal varices | 40 | 65703 | 7.2 (5.6, 8.7) | 99.2% | <0.001 |
| Esophageal stricture | 20 | 38594 | 1.2 (0.7,1.7) | 95.9% | <0.001 |
| GERD***** | 31 | 51389 | 9.6 (7.8, 11.4) | 98.7% | <0.001 |
| Bile reflux | 10 | 15178 | 4.5(2.9, 6.2) | 97.3% | <0.001 |
| Barret's Esophagus | 10 | 28671 | 0.9 (0.4,1.3) | 92.8% | <0.001 |
| Esophagitis | 42 | 65564 | 8.3 (7.1, 9.5) | 98.1% | <0.001 |
| Esophageal candidiasis | 29 | 49288 | 3.1 (2.3, 3.8) | 97.5% | <0.001 |
| Cardial varices | 4 | 7914 | 2.3(0.6, 4.1) | 95.9% | <0.001 |
| Achalasia | 16 | 23269 | 0.5 (0.03, 0.7) | 65.5% | <0.001 |
| Esophageal cancer | 35 | 74485 | 6.1 (5.3, 7) | 98.9% | <0.001 |
| Esophageal tumor | 10 | 20610 | 3.8(2, 5) | 97.5% | <0.001 |
| Hiatus Hernia | 33 | 56505 | 4.2 (3.5, 4.9) | 97.3% | <0.001 |
| Others | 37 | 47271 | 4.5 (3.8, 5.2) | 98.1% | <0.001 |

*Peptic ulcer disease, ** Gastric ulcer, *** Duodenal ulcer, **** Gastric outlet obstruction, *****Gastroesophageal Reflux Disease.

which showed relatively consistent prevalence across the regions, with pooled estimates ranging from 3% (2–3) in Western and Southern Africa to 4% (3–5) in Norther and Eastern Africa. Overtime, the prevalence of gastric cancer remained stable, with estimates of 3% in both 2000-2010 and 2011-2024.

Furthermore the analysis of clinical indications for endoscopy revealed notable regional variation and temporal trends across Africa. Starting with hematemesis, it was the most prevalent in North Africa, with a pooled prevalence of 23.7% (18.6-28.9), significantly higher than Eastern, Western, and Southern Africa. However, no significant changes in hematemesis prevalence were observed over time. Dyspepsia emerged as the most prevalent indication across all region, with the highest prevalence in Western Africa 62.6%(50.4-74.8). Despite leading indication, dyspepsia showed no significant change in prevalence over the periods.

GERD symptoms demonstrated regional and temporal trend difference, being significantly more prevalent in Norther Africa (17% (13–21), and Eastern Africa (19(12–25) compared to other regions. Overtime, the prevalence of GERD symptoms showed a significant increase, rising from 6% (4–8) during 2000-2010 to 15% (12–18) in the 2011-2024 period. Dysphagia, on the other hand, was more prevalent in Eastern Africa and Southern Africa, with pooled prevalence of 16% and 19% respectively. Unlike GERD symptoms, dysphagia did not exhibits significant temporal changes ibn prevalence (Table 4).

## Discussion

This systematic review and meta-analysis of 68 studies, covering 120,460 patients across Africa, provides an overview of UGIE indications and findings, offering new insights into the burden of UGID on the continent. One of the novel of this review is it the first, large-scale, continent-wide approach, with aggregated data from diverse region to better understand the prevalence, regional variation, and temporal trends of GI conditions. This review reveals that 78% of endoscopic evaluation detected organic findings, point out the high prevalence of UGID. Among the indication of UGIE, dyspepsia was identified as the most commonly reported indication for UGIE, with pooled prevalence of 52.4%. Other notable indications included, hematemesis (13.9%), and recurrent GERD symptoms (11.2%, alongside dysphagia (9.2%), vomiting (9.7%), and abdominal pain (17.4%). As for endoscopic findings, gastritis (33.3%) was the most frequent endoscopic finding, with PUD (15.1%) also being notable findings. The significance of this review lies in its ability to synthesize data from multiple

**Table 4.  Regional distribution, temporal trends, of endoscopic indications and findings.**

| Endoscopic findings | Pooled prevalence in African regions 95% CI | | | | Years of Trends | |
|---|---|---|---|---|---|---|
| | Western Africa | Northern Africa | Eastern Africa | Southern Africa | 2000-2010 | 2011-2024 |
| Esophageal cancer | 1(0, 1) | --------- | 10 (8, 13) | 10 (8, 12) | 6 (4, 7) | 7 (5, 8) |
| Esophagitis | 8 (5, 10) | 19 (11, 27) | 9 (7, 11) | 8 (7, 9) | 6 (4, 8) | 10 (8, 11) |
| Esophageal varices | 5 (3, 6) | 18 (7, 28) | 8(4, 12) | 6 (3, 9) | 6 (4,7) | 8 (5, 10) |
| Esophageal stricture | 1(0, 1) | 1(0, 1) | 0 (0, 1) | 3 (1, 4) | 2(1,4) | 1(0,1) |
| GERD | 9 (3, 15) | 18 (12, 24) | 9(6, 12) | 4 (1, 6) | 11(7, 14) | 9 (7, 12) |
| Esophageal candidiasis | 2 (1, 2) | 1 (0, 1) | 2(1, 2) | 6 (4, 9) | 4(2,5) | 2 (2,3) |
| Hiatal Hernia | 3 (2,5) | 9 (4,14) | 4(3,6) | 1(1,2) | 4(3,5) | 5 (4,7) |
| Gastric cancer | 3(2,3) | 4 (3,5) | 4(3,5) | 3(2,4) | 3(3,4) | 3 (3,4) |
| Gastritis | 33 (25, 40) | 45 (29, 60) | 29 (24,35) | 27 (14, 40) | 26 (19, 33) | 36 (30, 43) |
| Gastric polyp | 1(0, 1) | 1(0, 2) | 1(0, 1) | --------- | 1 (0, 1) | 1 (0, 1) |
| PUD | 19 (15, 23) | 12 (7,17) | 15(12,17) | 8(4, 12) | 15 (11, 18) | 15 (13, 18) |
| Gastric erosion | 15 (10, 20) | 9 (8,10) | 3 (0, 5) | 1(0, 1) | 2 (1,3) | 13 (9, 16) |
| DU | 11 (7, 14) | 15 (8, 22) | 11(8, 13) | 6 (4, 9) | 11 (8, 13) | 10 (8, 12) |
| Duodenitis | 11 (8, 14) | 27(2, 53) | 5 (4, 7) | 7(1, 13) | 12 (9, 15) | 10(7, 13) |
| **Indications** | **Pooled prevalence of indications in African regions 95% CI** | | | | **Years of Trends** | |
| Dysphagia | 3(2, 4) | 4 (3, 6) | 16 (11, 21) | 19 (12, 25) | 12(9, 14) | 9(7, 11) |
| GERD symptoms | 8 (6, 10) | 17 (13, 21) | 19 (12, 25) | 7 (2, 12) | 6 (4, 8) | 15 (12, 18) |
| Dyspepsia | 62.6(50.4, 74.8) | 43.5(35.8, 51.3) | 51.2(35.4, 67) | 35.2(21.5, 49) | 56.8(41.4, 72.1) | 52.3(42.1, 62.5) |
| Hematemesis | 11.1(8.9, 13.3) | 23.7(18.6, 28.8) | 11.8(6.3, 17.2) | 13.5(10.4, 16.7) | 12.5 (9.1, 15.8) | 14.5(11.8, 17.1) |
| Melena | 3.7(-0.4, 7.8) | 11(5.5, 16.5) | 4.6(2.1, 7.2) | 1.6(0.4, 2.8) | 6.6(-0.25, 15.7) | 6.1(3.7, 8.4) |
| Anemia | 1.2(0.7, 1.8) | 3.3(1.3, 5.4) | 1.3(-0.2, 2.8) | 2.5(1.4, 3.6) | 1.5 (1, 2) | 2.7(1.6, 3.9) |
| Vomiting | 4(2.6, 5.3) | 17.7(13.9, 21.4) | 13.6(9.4, 17.8) | 5.3 (1.1, 9.5) | 6.5 (4.6, 8.5) | 10.7(8.5, 12.9) |

countries. Importantly, this review identify key regional difference, such as higher prevalence of esophageal varices in Northern Africa and Eastern Africa and esophageal Cancer was in Eastern and Southern Africa, which can provide regional policies and improve diagnostic strategies. Furthermore, the inclusion of trends offers valuable insights into the changing patterns of UGID across African continent. We observed significance changes in the prevalence of condition like GERD over time, with increased rate

In our review, 52.4% of patients in Africa underwent UGIE due to dyspepsia, making it the most frequent indication for UGIE. Similarly, a study conducted in Asia reported that 48.3% UGIE were performed for dyspepsia [88]. These findings of dyspepsia as a common indication for UGIE point out significance indication in across different world, and as common indication across diverse region. Furthermore, the prevalence of dyspepsia were different across Africa, with Western Africa showing the highest prevalence (62.6%), followed by Eastern Africa (51.2%), and Northern Africa (43.5%). Southern Africa showed a comparatively lower prevalence at 35.2%. Importantly, the differences in the burden of dyspepsia between Western, Northern, and Eastern Africa were not statistically significant. However, a significant difference was observed between Western and Southern Africa, where the prevalence in Western Africa was notably higher than in Southern Africa. One possible explanation for this disparity may be, the highest H. pylori burden in countries with in Western Africa countries, reporting a prevalence of 87.7 [89], the highest ever recorded, as a well-known cause of dyspepsia. This review also observed temporal changes, with the dyspepsia burden slightly decreasing over the two decades studied, suggesting evolving factors influencing its prevalence. However, care should be used when interpreting the high prevalence found in our review. The patients in our analysis had concerning symptoms when they first arrived. They were examined by endoscopy in a GI clinic according to particular clinical indications. Therefore, it is possible that the pooled prevalence of dyspepsia were directly impacted by this selective process. Consequently, it's possible that the results don't accurately reflect the larger population.

Hematemesis, a key presentation of UGIB, had a prevalence of 13.9% across Africa, making it significant indication for UGIE, with substantial regional variation. Northern Africa 23.7% had the highest burden as compared to Western, Easter, and Southern Africa. This disparity may be linked to higher prevalence of esophageal varices (18%), esophagitis (19%), and GERD (18%) in Norther Africa. Over time, hematemesis prevalence rose slightly from 12.5% (2000-2010) to 14.5% (2011-2024), possible due to improved diagnostic access. This significant prevalence emphasize the need for early diagnosis and treatment. Patients also should be aware that rebleeding can happen in 7% to 16% of cases even after therapy [90].

Based on 35 studies out of 68 reviewed articles, we analyzed the burden of esophageal cancer and found a pooled prevalence of 6.1%. This prevalence exhibited regional variation, with higher burden of 10% in both Easter and Southern Africa, contrasting sharply with Western Africa, which reported a significantly lower prevalence of 1%. Interestingly, no significant trend change was observed over time, with the prevalence remaining relatively stable from 6% during 2000-2010 to 7% (2011-2024). This malignant tumor is the eighth most commonly diagnosed cancer and the sixth leading cause of cancer death worldwide [91], with a notably high burden in less developed regions, such as Africa, where nearly 80% of cases occur [92]. Global aging, population growth, and the prevalence of risk factors like the use of tobacco and alcohol, poor diet, inactivity, and obesity are all contributing to the rise in the incidence and mortality due to esophageal cancer [91–93]. It is highly malignant, and the outlook is frequently poor [94]. Considering the high incidence and dismal prognosis of esophageal cancer, especially in developing nations, we suggest starting and growing screening programs to identify the disease in its earlier, at its more curable stages. When it comes to high-risk areas, this is especially important. Furthermore, it is critical to raise public awareness of the risk factors

for esophageal cancer, which include unhealthy eating habits, lack of exercise, and alcohol and tobacco use.

Despite UGIE is not a gold standard for diagnosis of GERD, particularly for endoscopy-negative GERD [95], it is still a valuable diagnostic technique despite its drawbacks. Thirty one of the sixty-eight studies in our review used UGIE to diagnose GERD, revealed a pooled prevalence of 9.6% (7.8, 11.4), it is consistent with population-based reviews that were carried out in Australia, the Middle East, and East Asia, but lower than in North America (19.8%) and Europe (15.2%) [96]. Our reliance on endoscopic diagnosis, while population-based reviews used symptom-based diagnoses, could account for some of the discrepancies seen in these results.

Based on the review of the 59 publications, the PUD prevalence was 15.1% (13.2, 16.9), which is higher than the global review's 8.4% [97], and Canada 5.3% [98]. This disparity likely reflects the inclusion of patients referred for endoscopy due to severe or alarming symptoms, as per ASGE guidelines [10]. We must therefore exercise reservations when interpreting this pooled result because our review may have overestimated the prevalence of PUD. Nevertheless, our findings align with studies conducted in China (17%) [99] and Denmark 15.7% [100]. Regarding the regional distribution of PUD, it has variability between Western and Southern Africa, the Western showing the highest prevalence 15% (11–23), compared to lower prevalence of Southern Africa 8% (4–12). This disparity likely because of key risk factors such as H. pylori, NSAID use, and variation of the dietary habit between regions [101,102]. Despite the regional difference, the temporal trend of PUD prevalence across Africa appear stable, with 15% (11–18) during 2000-2010 and 15% (13–18) during 2011-2024. This stability indicates that the overall burden of PUD has not significantly changed over the last decades, may be because of risk factors may have not decrease overtime, and may not have substantial change in public health measures, such as widespread eradication of H. pylori or better treatment protocol.

The pooled prevalence of gastritis in our review, based on 53 published articles out of 68, was 33.3% (28.5, 38). While the regional variations are not statistically significant, Northern Africa share the highest burden (45% (29-60), with the Southern Africa had lowest prevalence 27% (14-40), with intermediate prevalence in Western 33% and Eastern Africa 29%. Over the 24 years period examined, the temporal trend indicates a noticeable increase in the prevalence of gastritis, from 2000-2010 (26%) to 36% during 2011-2024. Possible explanation for the high prevalence of gastritis in Africa could be the high H. pylori burden in the region [89,103–105], and use of non-steroidal anti-inflammatory drugs in local preparations [106]. Thus, addressing these factors critically important to minimizing complications such as gastric cancer, as point out by Professor Correa's, which describe the disease starts as normal gastric mucosa and progresses to chronic non-atrophic gastritis, chronic atrophic gastritis, intestinal metaplasia, and finally gastric cancer [107]. Prevention and effective management of gastritis can also reduce the strain of public health system [108].

Our review's pooled prevalence of gastric cancer was 3.3%, based on data from 43 studies. Comparatively, higher incidence are reported in East Asia, such as Korea, and Japan [109]. This discrepancy may stem from regional factors; in East Asia, high H. pylori burden [110,111], combined with dietary factors like high consumption of salt-preserved foods and dietary nitrite, act synergistically with H. pylori infection to increase risk and promote gastric cancer development [112]. Interestingly, despite H. pylori being endemic in Africa, the prevalence of gastric cancer remain relatively low, a phenomenon known as the "African enigma" [113]. This paradox suggest that although H. pylori infection is widespread, other factors such as such as dietary difference, genetic factors, or environmental influences, may affect the progression of gastric cancer.

Our review further reveal, the temporal trend has remained constant at 3% over the past two decades in Africa, however, the incidence of gastric cancer have fallen dramatically in US and elsewhere over the past several decades contrary to our view [112]. This constant prevalence over the past two decades in Africa may indicates that efforts to reduce risk factors have not yet introduced, and the trend may also reflect the slow nature of changes in cancer incidence, particularly for gastric cancer, which may take years or decades to show measurable effects. The regional distribution of gastric cancer in this review is evenly spread, ranging from 3-4%. It may suggest that similar factors to the regions may play a role in maintain a relatively such similar prevalence. Therefore, because gastric cancer is often associated with a poor prognosis, the main strategy for improving clinical outcomes is through primary prevention, the widespread introduction of refrigeration has led to a decrease in the intake of chemically preserved foods and increased consumption of fresh fruits and vegetables [114], improvements in sanitary and housing conditions, as well as the use of eradication therapy for H. pylori [115]

A wide, many and comprehensive data set and a focus on publications from the last 24 years, assessing the regional distribution and temporal trend variation which offer an updated overview of the indications and findings of endoscopy in Africa, are our review's notable strengths. Nevertheless, the review mainly uses institutional samples, therefore that findings at endoscopy may not be a true reflection of upper gastroduodenal diseases in Africa, this is because of frequent hospital visits have been linked to severe persistent complaints, alarming symptoms, or advanced age as candidate for UGIE. Furthermore, there is a great deal of study heterogeneity in our review, which could have an impact on the meta-analysis results. Notably, almost all of the studies that we analyzed had strong indications for endoscopy examinations, making it difficult to identify asymptomatic patients with upper gastrointestinal endoscopic findings. Moreover, this study did not account for H. pylori infection due to inconsistent reporting across the studies included in our analysis. As H. pylori is a key factor in several upper gastrointestinal diseases, future research should aim to address this gap by including data on H. pylori status to provide a more comprehensive understanding of its role. This review also focuses on the overall indication and findings of UGIE in Africa; as a result, we did not analyze age and sex-based differences in the indications and findings for UGIE, which may provide deep understandings demographic patterns, moreover the review does not explore in the relationship between specific indications, such as dyspepsia or hematemesis, and particular UGIE findings. Addressing these gaps in future research would enhance the understanding of the epidemiological and clinical variation in UGIE indications and findings in Africa

### Limitation of the study

The fact that our inclusion criteria were limited to research published in English may have resulted in the omission of important data from studies written in other languages. The comprehensiveness and generalizability of our findings may be impacted by this limitation.

### Conclusion

This systematic review and meta-analysis provide a comprehensive overview of UGIE indication and findings across Africa, revealing over 78% of abnormal findings. The most common indications with high prevalence were dyspepsia, abdominal pain, GERD symptom and hematemesis. Gastritis was the most common endoscopic findings, with notable regional variation, particularly Norther Africa, while PUD was with second most findings and more commonly associated with duodenal ulcer. The most common cancer observed by UGIE

was esophageal cancer, showed significant variation higher burden in Eastern and Southern Africa, while gastric cancer exhibited no significance regional and temporal differences. Finally, because many UGIE findings are often associated with a poor socio-economic status, the main strategy for improving clinical outcomes is through primary prevention, the widespread introduction of improvements in sanitary and housing conditions, as well as the use of eradication therapy for H. pylori.

## Supporting information

**S1 Checklist. PRISMA Checklist.** PRISMA 2020 checklist.
(DOCX)

**S1 Appendix. Search strategies.**
(DOCX)

**S2 Appendix. Quality assessment of included studies.**
(DOCX)

**S1 Fig. Depicts the schematic flow of study selection steps for indications and endoscopic findings of UGID in Africa.**
(DOCX)

**S2 Fig. Forest Plot of hematemesis as an Indication for Upper Gastrointestinal Endoscopy.**
(DOCX)

**S3 Fig. Forest Plot of GERD symptoms as an Indication for Upper Gastrointestinal Endoscopy (UGIE).**
(DOCX)

**S4 Fig. S3 Fig Forest Plot of Dysphagia as an Indication for Upper Gastrointestinal Endoscopy (UGIE).**
(DOCX)

**S5 Fig. Forest Plot of GERD as diagnosis of Upper Gastrointestinal Endoscopy (UGIE).**
(DOCX)

**S6 Fig. Forest Plot of Esophageal cancer as diagnosis of Upper Gastrointestinal Endoscopy (UGIE).**
(DOCX)

## Author contributions

**Conceptualization:** Seid Mohammed Abdu.

**Data curation:** Seid Mohammed Abdu, Ebrahim Msaye Assefa, Hussen Abdu.

**Formal analysis:** Seid Mohammed Abdu.

**Investigation:** Seid Mohammed Abdu, Ebrahim Msaye Assefa, Hussen Abdu.

**Methodology:** Seid Mohammed Abdu.

**Software:** Seid Mohammed Abdu.

**Supervision:** Seid Mohammed Abdu, Ebrahim Msaye Assefa, Hussen Abdu.

**Writing – original draft:** Seid Mohammed Abdu, Ebrahim Msaye Assefa, Hussen Abdu.

**Writing – review & editing:** Seid Mohammed Abdu, Ebrahim Msaye Assefa, Hussen Abdu.

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
