## [Decision Letter · Decision Letter 0]

23 Sep 2024

PONE-D-24-29678Indications and Endoscopic findings of upper gastrointestinal diseases in Africa: A systematic Review & Meta-analysisPLOS ONE

Dear Dr. Abdu,

Thank you for submitting your manuscript to PLOS ONE. After careful consideration, we feel that it has merit but does not fully meet PLOS ONE’s publication criteria as it currently stands. Therefore, we invite you to submit a revised version of the manuscript that addresses the points raised during the review process.

We look forward to receiving your revised manuscript.

Kind regards,

Muhammad Salman Bashir, M.S.C

Academic Editor

PLOS ONE

2. We note that your Data Availability Statement is currently as follows: [All relevant data are within the manuscript and its Supporting Information files]

3. As required by our policy on Data Availability, please ensure your manuscript or supplementary information includes the following:

Reviewers' comments:

Reviewer's Responses to Questions

**Comments to the Author**

1. Is the manuscript technically sound, and do the data support the conclusions?

Reviewer #1: No

Reviewer #2: Yes

Reviewer #3: Yes

Reviewer #4: No

2. Has the statistical analysis been performed appropriately and rigorously? 

Reviewer #1: I Don't Know

Reviewer #2: I Don't Know

Reviewer #3: Yes

Reviewer #4: No

3. Have the authors made all data underlying the findings in their manuscript fully available?

Reviewer #1: No

Reviewer #2: Yes

Reviewer #3: Yes

Reviewer #4: Yes

4. Is the manuscript presented in an intelligible fashion and written in standard English?

Reviewer #1: Yes

Reviewer #2: Yes

Reviewer #3: Yes

Reviewer #4: Yes

5. Review Comments to the Author

Reviewer #1: 1. Potential Factual Inaccuracy

The paper states that "to the best of our knowledge, there has been no systematic assessment of endoscopic indications and findings of upper GI disease in Africa." However, similar reports already exist, suggesting a potential factual inaccuracy. Specifically, the 2019 paper by Timothy N. Archampong et al., titled "Gastro-duodenal disease in Africa: Literature review and clinical data from Accra, Ghana," provides data on the prevalence of PUD and H. pylori infection. This report emphasizes the need to consider H. pylori in the current study's analysis, particularly because NSAID use was identified as a possible reason for the persistent high prevalence of PUD in some African countries.

2. Limitations in Search Methodology

Despite the search period extending up to 2024, important recent studies are missing from the analysis. For example, the 2023 study titled "Indications and Findings of Upper Gastrointestinal Endoscopy at a Tertiary Hospital in Ethiopia: A Cross-Sectional Study" was not included in the results. This raises concerns about the quality of the search strategy or hand search, suggesting that other significant papers may also have been overlooked.

3. Inadequate Search Results

The search string provided in the paper only resulted in finding an older study (a 1994 paper on prescriptions for peptic ulcer drugs in Iceland), which suggests a problem with the effectiveness of the search strategy. This raises concerns about whether the search strategy was able to capture a comprehensive set of relevant literature.

4. Lack of Data Availability

The paper mentions that "all relevant data are within the manuscript and its Supporting Information files," but no data or links were provided, making it impossible to verify the findings. This lack of data transparency could impact the reproducibility and credibility of the study.

5. Concerns Regarding Data Bias and Representativeness

The exclusion of data in native languages may have introduced bias in the analysis. Additionally, the paper does not provide information on the overall number of upper GI endoscopy cases in Africa or how the current report represents a percentage of that total, raising concerns about the representativeness of the study. Specific data on these points would help in assessing the study's generalizability.

Reviewer #2: The researchers conducted a systematic review and meta-analysis entitled “Indications and Endoscopic findings of upper gastrointestinal diseases in Africa: A systematic Review & Meta-analysis

The review and meta-analysis are well designed and conducted. The manuscript is well written.

However, several points could be improved in the manuscript.

1. The author reported as primary indication:

epigastric pain (50%), dyspepsia (43.7%), abdominal pain (29.6%),

only these 3 indications reached 123.3%? please clarify and explain the sum of more than 100%.

Moreover, epigastric pain, dyspepsia and abdominal pain could used for upper abdominal pain,

Please explain the differences between the different indication as used in the studies and ensure that more than one of these indications not used in the same patients (maybe the patients had dyspepsia and abdominal pain as indication)

2. Functional dyspepsia affected 29.8% of patients,

Please explain how wad functional dyspepsia was diagnosed in EGD. Functional dyspepsia is diagnosed according to clinical criteria and not according EGD.

3. Table 1: Reference number 50: please add the name of the author

4. Abbreviations: should explained in the first appearance in the text and below the tables, for example table 3.

5. What about Helicobacter pylori, is there any data in the studies? If not, please add a note in the limitations.

6. Much of the discussion seems to repeat the results. The focus should be on discussing the findings. Why are the findings in Africa different from other regions in the world? Consider lifestyle, diet, chronic diseases like diabetes and obesity...and other causes. For example, reflux is lower in Africa.

I would suggest including more studies from the Middle East, which is close to Africa, if available. Here are three examples:

Abu-Freha N, Gat R, Philip A, Yousef B, Ben Shoshan L, Yardeni D, Nevo-Shor A, Novack V, Etzion O. Indications and Findings of Upper Endoscopies in Males and Females, Are They the Same or Different? J Clin Med. 2021 Apr 11;10(8):1620. doi: 10.3390/jcm10081620. PMID: 33920408; PMCID: PMC8070302.

Sperber AD, Bangdiwala SI, Drossman DA, Ghoshal UC, Simren M, Tack J, Whitehead WE, Dumitrascu DL, Fang X, Fukudo S, Kellow J, Okeke E, Quigley EMM, Schmulson M, Whorwell P, Archampong T, Adibi P, Andresen V, Benninga MA, Bonaz B, Bor S, Fernandez LB, Choi SC, Corazziari ES, Francisconi C, Hani A, Lazebnik L, Lee YY, Mulak A, Rahman MM, Santos J, Setshedi M, Syam AF, Vanner S, Wong RK, Lopez-Colombo A, Costa V, Dickman R, Kanazawa M, Keshteli AH, Khatun R, Maleki I, Poitras P, Pratap N, Stefanyuk O, Thomson S, Zeevenhooven J, Palsson OS. Worldwide Prevalence and Burden of Functional Gastrointestinal Disorders, Results of Rome Foundation Global Study. Gastroenterology. 2021 Jan;160(1):99-114.e3. doi: 10.1053/j.gastro.2020.04.014. Epub 2020 Apr 12. PMID: 32294476.

Sperber AD, Freud T, Abu-Freha N, Shibli F, Brun R, Bangdiwala SI, Palsson OS, Dickman R. Epidemiology of disorders of Gut-Brain interaction in Israel: Results from the Rome Foundation global epidemiology study. Neurogastroenterol Motil. 2022 Aug;34(8):e14323. doi: 10.1111/nmo.14323. Epub 2022 Jan 24. PMID: 35072332.

Reviewer #3: This is an important article providing valuable information on gastrointestinal diseases in Africa, through a systematic review and meta-analysis conducted using electronic databases such as PubMed, Google Scholar and Hinari, and manual searches of relevant websites.

I believe that your article, if published, will make positive contributions to the medical literature.

Reviewer #4: The scope of data taken is too broad, so the results are not specific. There are overlapping indications, for example, dyspepsia, early satiety, anorexia, bloating, postprandial fullness, vomiting, and belching. I evaluate this article that there is nothing we can learn from this paper.

6. PLOS authors have the option to publish the peer review history of their article (what does this mean? ). If published, this will include your full peer review and any attached files.

**Do you want your identity to be public for this peer review?** For information about this choice, including consent withdrawal, please see our Privacy Policy .

Reviewer #1: No

Reviewer #2: No

Reviewer #3: No

Reviewer #4: No

---

## [Author Response · Author response to Decision Letter 1]

16 Oct 2024

A Rebuttal Letter

Dear esteemed editor Dr. Muhammad Salman Bashir and reviewers,

We greatly value your feedback and comments provided on our manuscript titled "Indications and Endoscopic Findings of Upper Gastrointestinal Diseases in Africa: A Systematic Review & Meta-analysis". All of the points that you raised are important, improve the quality of the work, and contain details that the journal requires.

Dear, we made every effort to answer every one of your remarks. The responses that follow have been made in response to each one. Please find below a point-by-point response to all comments and concerns raised. Reviewer comments are shown in black, followed by our corresponding responses in dark blue. Changes to the manuscript have been highlighted in blue in the revised version for easy access. We want to thank you again for your wonderful feedback and appreciate any more suggestions or remarks you may have to help us improve the work.

Best regards,

A point-by-point response to the comments given

A. Response to Comments of the Academic Editor

Comment #1: Dear Dr. Abdu,

Thank you for submitting your manuscript to PLOS ONE. After careful consideration, we feel that it has merit but does not fully meet PLOS ONE’s publication criteria as it currently stands. Therefore, we invite you to submit a revised version of the manuscript that addresses the points raised in the review process.

If applicable, we recommend that you deposit your laboratory protocols in protocols.io to enhance the reproducibility of your results.

Response #1: Dear editor, we agree with your suggestion and we have tried to incorporate all of your concerns throughout the revised manuscript. We also tried to adjust the manuscript to follow the requirements of PLOS ONE's style.

Comment #2. We note that your Data Availability Statement is currently as follows: [All relevant data are within the manuscript and its Supporting Information files]

Please confirm at this time whether or not your submission contains all the raw data required to replicate the results of your study. Authors must share the “minimal data set” for their submission. PLOS defines the minimal data set to consist of the data required to replicate all study findings reported in the article, as well as related metadata and methods. For example, authors should submit the following data:

- The values behind the means, standard deviations, and other measures reported;

Authors do not need to submit their entire data set if only a portion of the data was used in the reported study. If your submission does not contain these data, please either upload them as Supporting Information files or deposit them to a stable, public repository and provide us with the relevant URLs, DOIs, or accession numbers.

Response #2: Thank you for your query regarding the Data Availability Statement. Since our study is a systematic review and meta-analysis, the data used for the analysis are publicly available and were extracted from previously published studies. Therefore, all the relevant data are already available within the manuscript and its Supporting Information files.

No additional raw data were generated or analyzed specifically for this study, and therefore, the minimal data set consists of the extracted data from these published studies, which are fully documented and cited in the manuscript. Furthermore, we have included sample forest plots in the Supporting Information section of this revision, which displays the results presented in the tables. So, we believe all necessary data required to replicate our study findings are available within the manuscript and Supporting Information. If further clarification or additional information is required, please let us know.

Comment #3: If there are ethical or legal restrictions on sharing a de-identified data set, please explain them in detail (e.g., data contain potentially sensitive information, data are owned by a third-party organization, etc.) and who has imposed them (e.g., an ethics committee). Please also provide contact information for a data access committee, ethics committee, or other institutional body to which data requests may be sent. If data are owned by a third party, please indicate how others may request data access.

Response #3: Our study is a systematic review and meta-analysis, and all data included in the analysis were derived from publicly available sources, specifically, previously published studies. As no primary data were collected, there are no ethical or legal restrictions on sharing a de-identified data set. The data used are already in the public domain, and thus, no specific ethical approval or restrictions apply to the sharing of this data.

Comment #4: As required by our policy on Data Availability, please ensure your manuscript or supplementary information includes the following: A numbered table of all studies identified in the literature search, including those that were excluded from the analyses. For every excluded study, the table should list the reason(s) for exclusion.

Response #4: Dear, we greatly appreciate your comment regarding the presentation of excluded studies. We value the importance of transparency in the selection process. However, given that our literature search identified 2,850 studies, it is impractical to include a numbered table listing each study and the reasons for exclusion. Instead, we have provided a comprehensive overview of the exclusion criteria in Figure 1 of the manuscript. This figure outlines the primary reasons for exclusion, such as duplicate records, studies not conducted in Africa, reviews or editorials, and studies that did not report the outcome of interest, among others. We believe this presentation effectively summarizes the reasons for exclusion and ensures clarity in our study selection process. If you require additional details or clarification, please let us know, and we will be happy to provide further information.

Comment #5: If any of the included studies are unpublished, include a link (URL) to the primary source or detailed information about how the content can be accessed.

Response #5: Thank you for your query. We would like to clarify that no unpublished studies were included in our systematic review and meta-analysis.

Comment #6: A table of all data extracted from the primary research sources for the systematic review and/or meta-analysis. The table must include the following information for each study: Name of data extractors and date of data extraction. Confirmation that the study was eligible to be included in the review. All data extracted from each study for the reported systematic review and/or meta-analysis that would be needed to replicate your analyses.

Response #6: Thank you for your comment regarding the data extraction table. As requested, we have included a detailed table in the Supporting Information file of the revised manuscript.

Comment #7: If data or supporting information were obtained from another source (e.g. correspondence with the author of the original research article), please provide the source of data and dates on which the data/information were obtained by your research group.

Response #7: Thank you for your concern regarding the source of data and supporting information. For our systematic review and meta-analysis, all data were extracted directly from the original published studies included in our analysis. We did not obtain any additional data or supporting information from other sources, such as correspondence with the authors of the original research articles. If any data had been obtained through direct communication with the authors, we would have provided the appropriate sources and dates in the manuscript. However, this did not apply to our study.

Comment #8: If applicable for your analysis, a table showing the completed risk of bias and quality/certainty assessments for each study or outcome. Please ensure this is provided for each domain or parameter assessed. For example, if you used the Cochrane risk-of-bias tool for randomized trials, provide answers to each of the signaling questions for each study. If you used GRADE to assess the certainty of the evidence, provide judgments about each of the quality of evidence factors. This should be provided for each outcome.

Response #8: Thank you for your comment regarding the risk of bias and quality/certainty assessments. For our systematic review and meta-analysis, we assessed the risk of bias and quality of included studies using the Joanna Briggs Institute (JBI) tool, as appropriate for the types of studies analyzed. We have provided a table in the Supporting Information file that includes the completed risk of bias and quality assessments for each study, covering all relevant domains and parameters assessed using the JBI tool. This table contains specific details for each domain and parameter, ensuring full transparency in the evaluation of each study.

Comment #9: An explanation of how missing data were handled.

Response #9: Thank you for your comment regarding the handling of missing data. In our systematic review and meta-analysis, we focused on upper gastrointestinal (UGI) endoscopy indications reported in at least two or more studies. If a study did not report a particular indication, we interpreted this as a potential absence of data for that specific indication, rather than as missing data. To ensure robust analyses, we included only the indications that were consistently reported in two or more studies. If a study lacked data on a particular indication, it was excluded from that specific analysis but still included in others where it met the reporting criteria. This approach allowed us to focus on well-reported indications while maintaining the integrity of our meta-analysis.

B. Response to Comments of the Reviewers

Review Comments to the Author

Response to Comments of the Reviewer #1

Comment #1: Potential Factual Inaccuracy. The paper states that "to the best of our knowledge, there has been no systematic assessment of endoscopic indications and findings of upper GI disease in Africa." However, similar reports already exist, suggesting a potential factual inaccuracy. Specifically, the 2019 paper by Timothy N. Archampong et al., titled "Gastro-duodenal disease in Africa: Literature review and clinical data from Accra, Ghana," provides data on the prevalence of PUD and H. pylori infection. This report emphasizes the need to consider H. pylori in the current study's analysis, particularly because NSAID use was identified as a possible reason for the persistently high prevalence of PUD in some African countries.

Response #1: Thank you for raising this concern. We would like to clarify that there is no factual inaccuracy in the statement. The paper by Timothy N. Archampong et al. (2019) is a literature review that focuses primarily on peptic ulcer disease (PUD), its pattern, and gastric histopathology across Africa. Their study does not provide a systematic review and meta-analysis of all endoscopic indications and findings, it narrows its focus to peptic ulcer disease (PUD) and related histopathological features. They included 9 studies from both Africa and outside Africa to discuss PUD and its patterns, and 17 studies to explore the histopathology of PUD. In contrast, our study is a systematic review and meta-analysis that aims to provide a comprehensive evaluation of the most common indications for upper gastrointestinal endoscopy (UGIE) and to summarize the overall endoscopic findings related to upper gastrointestinal diseases (UGID) across Africa only. We synthesized data from 48 studies, identifying a wide range of indications (such as epigastric pain and dyspepsia) and endoscopic findings (such as gastritis, functional dyspepsia, and GERD). Our work offers a broader and more systematic assessment of UGID across the continent only, making it distinct from the literature review by Archampong et al. Therefore, there is no factual inaccuracy in our statement, as our study represents the first comprehensive systematic review and meta-analysis on this specific topic.

Comment #2: Limitations in Search Methodology

Despite the search period extending up to 2024, important recent studies are missing from the analysis. For example, the 2023 study titled "Indications and Findings of Upper Gastrointestinal Endoscopy at a Tertiary Hospital in Ethiopia: A Cross-Sectional Study" was not included in the results. This raises concerns about the quality of the search strategy or hand search, suggesting that other significant papers may also have been overlooked.

Response #2: Thank you for your valuable feedback. We acknowledge that the 2023 study titled "Indications and Findings of Upper Gastrointestinal Endoscopy at a Tertiary Hospital in Ethiopia: A Cross-Sectional Study" was inadvertently omitted from Table 1 in our manuscript. However, the study findings were indeed included in our analysis. We have now corrected this oversight by updating Table 1 to ensure that the study is properly listed among the included articles. We apologize for this error and have revised the table to accurately reflect the studies used in our analysis.

Upon further review of the included studies, we discovered that a study from Burkina Faso, conducted by Coulibaly et al., was inadvertently included in our presentation despite not meeting our inclusion criteria. While this study was excluded from our analysis, it was mistakenly listed in Table 1, taking the position of the Ethiopian study. The Ethiopian study was included in the analysis, but due to this oversight, it was not reflected correctly in the table. We have now corrected this error by removing the Burkina Faso study from the table and reinstating the Ethiopian study in its rightful position. Consequently, the total number of studies included remains 48, and the findings from the Ethiopian study have been fully accounted for in the analysis.

We appreciate the reviewer’s concern regarding the thoroughness of our search strategy. We would like to emphasize that we conducted a comprehensive search and hand search to capture all relevant studies. The issue with the Ethiopian study was not a failure of the search strategy but an inadvertent omission from the table. We have now rectified this, and we will conduct additional verification to ensure that no further significant studies were overlooked.

Comment #3: Inadequate Search Results

The search string provided in the paper only resulted in finding an older study (a 1994 paper on prescriptions for peptic ulcer drugs in Iceland), which suggests a problem with the effectiveness of the search strategy. This raises concerns about whether the search strategy was able to capture a comprehensive set of relevant literature.

Response #3: Thank you for your valuable feedback. We acknowledge your concern regarding the search strate

---

## [Decision Letter · Decision Letter 1]

17 Dec 2024

PONE-D-24-29678R1Indications and Endoscopic findings of upper gastrointestinal diseases in Africa: A systematic Review & Meta-analysisPLOS ONE

Dear Dr. Abdu,

Thank you for submitting your manuscript to PLOS ONE. After careful consideration, we feel that it has merit but does not fully meet PLOS ONE’s publication criteria as it currently stands. Therefore, we invite you to submit a revised version of the manuscript that addresses the points raised during the review process.

We look forward to receiving your revised manuscript.

Kind regards,

Muhammad Salman Bashir, M.S.C

Academic Editor

PLOS ONE

Reviewers' comments:

Reviewer's Responses to Questions

**Comments to the Author**

1. If the authors have adequately addressed your comments raised in a previous round of review and you feel that this manuscript is now acceptable for publication, you may indicate that here to bypass the “Comments to the Author” section, enter your conflict of interest statement in the “Confidential to Editor” section, and submit your "Accept" recommendation.

Reviewer #3: All comments have been addressed

Reviewer #5: All comments have been addressed

Reviewer #6: (No Response)

Reviewer #7: (No Response)

2. Is the manuscript technically sound, and do the data support the conclusions?

Reviewer #3: Yes

Reviewer #5: Yes

Reviewer #6: Yes

Reviewer #7: Partly

3. Has the statistical analysis been performed appropriately and rigorously? 

Reviewer #3: Yes

Reviewer #5: Yes

Reviewer #6: Yes

Reviewer #7: I Don't Know

4. Have the authors made all data underlying the findings in their manuscript fully available?

Reviewer #3: Yes

Reviewer #5: Yes

Reviewer #6: Yes

Reviewer #7: Yes

5. Is the manuscript presented in an intelligible fashion and written in standard English?

Reviewer #3: Yes

Reviewer #5: Yes

Reviewer #6: Yes

Reviewer #7: No

6. Review Comments to the Author

**Reviewer #3:**  Dear Author,

I have previously evaluated the article titled "Indications and Endoscopic findings of upper gastrointestinal diseases in Africa: A systematic Review & Meta-analysis" with the number PONE-D-24-29678R1 and stated that its publication in your journal would contribute positively to the medical literature. I have also reviewed the criticisms of other referees and the authors' responses to the criticisms regarding the article sent to me for re-evaluation. I believe that all the corrections made by the authors after the criticisms of other referees are sufficient. In my opinion, the article would be appropriate to publish.

Best regards.

**Reviewer #5: ** The authors have responded adequately to the review comments, addressing most of the concerns thoroughly. While there remains an unresolved point regarding the lack of specific data on the overall number of upper GI endoscopy cases in Africa and the study's proportionate representation within that total (Comment #6), the authors have explained that a comprehensive data source for all UGIE cases across Africa is unavailable. Given this limitation, the authors have taken steps to mitigate representational bias by including a geographically diverse range of studies and acknowledging this limitation in the revised manuscript.

The remaining unresolved point is noted as a future consideration and does not detract from the overall adequacy of the authors' responses. The reviewers find the authors' explanations and the additional transparency measures, including the full search strategy and supporting data files, sufficient to support the credibility and generalizability of the study's findings.

**Reviewer #6:**  I have been included as a reviewer after some comprehensive recommendations were made by others and read the latest corrected text. This systematic review and meta-analysis, while comprehensive, does have limitations that may affect the generalizability of the findings. 1) The reliance on institutional samples could lead to an overestimation of the prevalence of UGIE indications and findings, as these samples often consist of patients presenting with severe, persistent symptoms or advanced age. 2) The potential for publication bias should also be noted; studies reporting significant or positive results are more likely to be published, which may skew the pooled prevalence rates reported in the analysis. 3)Additionally, the restriction to English-language publications may have resulted in the exclusion of relevant data from non-English studies, further affecting the comprehensiveness of the findings. 4)Moreover, Africa is a huge continent where life style, socioeconomics or eating habits varies from north to south + east to west. Instead of generalisation of the whole continent data, it would be better if authors would be able to focus on regional findings. However, the solution of this problem will bring some workload as further dataset analysis that the authors may not prefer. 5) The discussion addresses the frequency of all symptoms and outcomes and compares them as numbers. However, does not discuss substantially the possible causes that those differences arise.

**Reviewer #7:**  Reviewer Comments for: Indications and Endoscopic Findings of Upper Gastrointestinal Diseases in Africa – A systematic Review and Meta-analysis - PONE-D-24-29678R1

1. The abstract should be re-written based on changes suggested below.

2. The introduction section does not adequately explain why it was important to review published reports of indications for, and findings at upper gastrointestinal endoscopy (UGIE) in Africa over a 20-year period. Merely reviewing the pooled prevalence of indications and endoscopic findings in Africa, as the authors have done, is, in my opinion, not enough. Key issues to bring forward here are that these indications/findings may vary across the continent, may change over time and may differ according to age and gender. The main purpose of the review, then, should be to help further understanding of upper gastroduodenal diseases in Africa and to see how they compare to the rest of the world.

3. Methods:

a) The authors should define what they mean by “indications” and “findings” based on standard criteria. The sentence on line 31 and 33 lists indications for UGIE but is inappropriately referenced: Ref 7 (Almario et al., 2018) does not even mention the word “endoscopy” in the entire manuscript while ref 10 is an incomplete book reference. A more suitable recent reference for indications for UGIE would be this: ASGE Standards of Practice Committee et al. Appropriate use of GI endoscopy. Gastrointest Endosc 2012; 75: 1127 – 31. doi: 10.1016/j.gie.2012.01.011

b) Indications for UGIE appeared not to have been searched enough: only specific findings relating to UGI tract - PUD/Gastroduodenal ulcers/ulcer - appeared to have been searched for. The other possible findings relating to the oesophagus, stomach and duodenum were not searched for at all. (See S1 Appendix)

c) Many papers, published from Africa, about indications and findings of UGIE were not included in the review. A simple Google Scholar search - indications and upper gastrointestinal endoscopy findings in Africa - will pull up many, many relevant papers, from all regions of the continent and which fulfilled the stated inclusion criteria, were not included in the review . Here's a link of my search: https://scholar.google.com/scholar?as_ylo=2020&q=Indications+and+upper+gastrointestinal+endoscopy+findings+in+Africa&hl=en&as_sdt=0,5 A simple similar PUBMED search also brought up many African articles not included in the review: https://pubmed.ncbi.nlm.nih.gov/?term=indications+and+findings+for+upper+gastrointestinal+endoscopy+in+Africa&size=200&filter=dates.2000%2F1%2F1-2024%2F4%2F1

d) Inclusion criteria (line 69): what specific months between 2000 and 2024 was the review based on?

e) Result (Line 93): Caption should be Results not Result

f) Line 97 states 45 publications, instead of 48 included in the analysis

g) Under characteristics of studies (line 106 to 108), the arithmetic does not add up: 18+13+8+8 = 47 NOT 48

h) Line 108 – concerning studies conducted in West Africa: 9+5=1= 15 NOT 18. Cameroon and Democratic Republic of Congo are considered Central African countries, not West African.

i) Line 110: the authors state: “….additionally, the criteria used to identify UGID were consistent across studies (Table 1)” I do not see any criteria listed in table 1 or any reference provided.

4. Comments on Table 1:

a) Countries in the same region not grouped together. This would have been easier on the eye and more logical

b) Author-name use inconsistent: In many, surnames and initial(s) are listed while in others, initials come before surnames, while in some, full names are given. In a few listings, only the surname is provided or the first name is given in full, with the surname given as an initial.

c) F Ankouane et al. (2015) appears twice and is referenced as nos. 53 and 54. In the reference list, however, 53 is given as Andoulo et al. (2015) and Ankouane et al. (2016). Can the authors resolve this?

d) A column showing percent male (or female) will be helpful in showing gender differences across studies being reviewed.

e) Most of the studies reviewed were retrospective. It will be nice to show period (years) spanning each study (eg. Ayuo et al.’s study spans 1993 – 2003).

5. Indications for Upper Gastrointestinal endoscopy (line 114):

a. Epigastric pain and dyspepsia are used as separate indications. According to Rome IV criteria, epigastric pain is part of dyspepsia (https://theromefoundation.org/rome-iv/rome-iv-criteria/).

b. Most studies on indications and findings of UGI in Africa do not categorize epigastric pain and dyspepsia as separate entities. Additionally, according to the World Endoscopy Organization, dyspepsia is the preferred standard term to refer to symptoms of upper GI disease, which may include epigastric pain, as a reason (indication) for endoscopy. Please see: https://www.worldendo.org/resources/minimal-standard-terminology

c. The two conditions should be treated together.

6. Comment on Table 2.

a. GERD symptoms (third row) are many - including heartburn and regurgitation. Which are the authors referring to: symptoms of GERD or GERD diagnosis? Heartburn is also listed an indication in the table. Indications (reasons) for endoscopy may be symptoms or diseases. Please see: https://www.worldendo.org/resources/minimal-standard-terminology

b. Some of the information given is perplexing to me, and I wonder if these are accurate: for example, the indication to rule out cancer of the stomach was reported in only two studies among 4,670 patients. Or take dyspepsia, as an indication: Only about a third of papers (15/48) reported on this indication in about a third of patients (30,105/100,710).

c. Can the authors also show pooled prevalence over time periods, for example, 2000 – 2010 and 2011 – 2024 to see any time trend?

7. Endoscopic Findings on Upper Gastrointestinal Tract (line 124):

a. Functional dyspepsia is not "a finding" at endoscopy - According to Rome IV criteria, it's the absence of structural disease to explain symptoms. If the authors mean no abnormal finding, they should say so.

b. Result for PUD should also give a breakdown of gastric ulcer (GU), duodenal ulcer (DU), or both in the text for emphasis as such is an important part of the review.

8. Comment on Table 3.

a. Port hepatic gastropathy should be portal hypertensive gastropathy

b. Functional dyspepsia is not an endoscopic finding. Authors to please see the reviewer’s comment above.

c. Bile reflex should be bile reflux

d. It will be very informative to provide a column comparing pooled prevalence of endoscopic findings during specific periods of the study. For example, what are the findings during the first part (2000 to 2010) and the later part (2011 - 2024) of the review? This way, readers can see if there’s any change in trend in findings, which the authors should address in the discussion section.

9. Discussion:

a. General comments:

i. Discussion of the significance of the results of the systematic review inadequate.

ii. Since this is an African review, I have not seen how the findings have been compared across Africa to see any similarities or differences. Example: is dyspepsia as common indication in East Africa as in West Africa? What are the most common endoscopic findings in each African region? Are they similar or different, and why?

iii. What are the age and gender similarities and differences in indications and findings of upper GI endoscopy in Africa?

b. The sentence on lines 144 and 145 states: “This result ranked higher than a population-based study of epigastric pain(21%) conducted in the USA, Canada, and the UK[66].” I do not think this is a valid comparison. Aziz et al.'s study is about a subtype of functional upper gastrointestinal disorder called epigastric pain syndrome, not epigastric pain in general, which may be associated with other upper gastrointestinal diseases including GERD, PUD, gastritis etc.

c. I don't think the authors have adequately discussed the significance of epigastric pain/dyspepsia as an indication (reason for endoscopy) especially in relation to findings at endoscopy. Is epigastric pain/dyspepsia more likely to be associated with any particular upper gastrointestinal finding at endoscopy?

d. Authors state in line 154 that “the wide definitions of dyspepsia in the included studies are probably the cause of the high prevalence in our review”. This statement is incorrect to me. Patients complained of symptoms of upper gastrointestinal disease which, collectively, are referred to as dyspepsia. All it means is that it's a common symptom and is not related to any definition.

e. Concerning upper gastrointestinal bleeding (line 156), the authors have not adequately discussed the significance of this observation especially as it relates to findings at UGIE. What are the most common findings in patients presenting with hematemesis and melena? Are there age, gender or regional differences in the prevalence? What do these mean?

f. Reference 67 (Ford et al., 2015) quoted on line 157 (at the end of the sentence on upper gastrointestinal bleeding) is about global prevalence of uninvestigated dyspepsia, nothing to do UGI bleeding mortality.

g. Concerning “functional dyspepsia” (line 173): functional dyspepsia in not an endoscopic finding. If the authors are referring to normal findings at endoscopy, they should say so. They cannot compare this with the global or regional prevalence of functional dyspepsia as defined by Rome IV criteria. Please see: https://theromefoundation.org/rome-iv/rome-iv-criteria/

h. Concerning line 181 – PUD prevalence: the authors have not satisfactorily discussed the significance of their finding. Has there been any change in the pattern of endoscopic finding of PUD (GU and DU) in Africa in the period under a review? What might be responsible for this?

i. Concerning line 186, the prevalence of gastritis: what is the prevalence for gastritis in various regions of Africa according to this review? Is there a change in prevalence of finding of gastritis over 20 years? What is the likely cause? What does it mean?

j. Concerning line 187, the putative high burden of H pylori in Africa: What is this high burden of Helicobacter pylori infection across Africa? How is it different in various African sub-regions? Is H pylori infection rising or declining or stable?

k. Concerning line 192, the authors should discuss regional African differences of gastric cancer and how this pooled prevalence compares to the rest of the world.

l. Concerning lines 193 – 201, I really don’t understand the point(s) the authors are making. Their review is about indications (reasons why people are requested to do endoscopy) and findings (what the examination discovers). Endoscopies are in Africa are usually done in hospitals (“institutions”), so I don’t see how reviewing this information overstates anything. If the authors are implying that findings at endoscopy are not a true reflection of upper gastroduodenal diseases in Africa, they should say so.

10. Conclusion: This should be re-rewritten after all the issues brought up above have been fully addressed.

11. References:

a. Referencing style of journal not adhered to - references with multiple (presumably more than 6) authors are merely given as first author, et al.

b. Some references not complete – see Ref 10 – Long DL, Harrison’s principles of internal medicine. 2012.

c. Many of the references are not appropriate for the claim being made. I have indicated some in my comments above.

7. PLOS authors have the option to publish the peer review history of their article (what does this mean? ). If published, this will include your full peer review and any attached files.

**Do you want your identity to be public for this peer review?** For information about this choice, including consent withdrawal, please see our Privacy Policy .

Reviewer #3: No

Reviewer #5: **Yes: ** Hiroshi Mihara

Reviewer #6: No

Reviewer #7: No

---

## [Author Response · Author response to Decision Letter 2]

26 Jan 2025

A Rebuttal Letter

Dear esteemed editor Dr. Muhammad Salman Bashir and reviewers,

We greatly value your feedback and comments provided on our manuscript titled "Indications and Endoscopic Findings of Upper Gastrointestinal Diseases in Africa: A Systematic Review & Meta-analysis". All of the points that you raised are important, improve the quality of the work, and contain details that the journal requires.

Dear, we made every effort to answer every one of your remarks. The responses that follow have been made in response to each one. Please find below a point-by-point response to all comments and concerns raised. Reviewer comments are shown in black, followed by our corresponding responses in dark blue. Changes to the manuscript have been highlighted in blue in the revised version for easy access. We want to thank you again for your wonderful feedback and appreciate any more suggestions or remarks you may have to help us improve the work.

Best regards,

A point-by-point response to the comments given

A. Response to Comments of the Academic Editor

Dear Dr. Abdu,

Thank you for submitting your manuscript to PLOS ONE. After careful consideration, we feel that it has merit but does not fully meet PLOS ONE’s publication criteria as it currently stands. Therefore, we invite you to submit a revised version of the manuscript that addresses the points raised during the review process.

We look forward to receiving your revised manuscript.

Kind regards,

Muhammad Salman Bashir, M.S.C

Academic Editor

PLOS ONE

Reviewers' comments:

Reviewer's Responses to Questions

Comments to the Author

1. If the authors have adequately addressed your comments raised in a previous round of review and you feel that this manuscript is now acceptable for publication, you may indicate that here to bypass the “Comments to the Author” section, enter your conflict of interest statement in the “Confidential to Editor” section, and submit your "Accept" recommendation.

Reviewer #3: All comments have been addressed

Reviewer #5: All comments have been addressed

Reviewer #6: (No Response)

Reviewer #7: (No Response)

2. Is the manuscript technically sound, and do the data support the conclusions?

Reviewer #3: Yes

Reviewer #5: Yes

Reviewer #6: Yes

Reviewer #7: Partly

Response: Thank you for your comments on the manuscript. I appreciate your insights and recognize the importance of ensuring that the research is technically sound and that the data fully supports the conclusions. I have addressed the points you raised below to clarify and strengthen the manuscript:

3. Has the statistical analysis been performed appropriately and rigorously?

Reviewer #3: Yes

Reviewer #5: Yes

Reviewer #6: Yes

Response: Reviewers #3, #5, and #6. Thank you for your positive assessment regarding the statistical analysis. I appreciate your acknowledgment of the rigor and appropriateness of the methods employed in this study.

Reviewer #7: I don’t Know

Response: Thank you for your comment regarding the statistical analysis. I understand the importance of ensuring that the methods used are clearly described and rigorously applied. The statistical analysis was conducted using Stata version 17, a reliable and widely used tool for meta-analysis and statistical modeling. Forest plots were generated using Stata as supporting information to visually present the pooled estimates and heterogeneity across studies. Additionally, the extracted dataset was provided to the journal as supplementary material for transparency and to facilitate reproducibility.

4. Have the authors made all data underlying the findings in their manuscript fully available?

Reviewer #3: Yes

Reviewer #5: Yes

Reviewer #6: Yes

Reviewer #7: Yes

Response: Reviewers #3, #5, #6, and #7 Thank you for your positive assessment regarding the data availability in our manuscript. We are pleased to confirm that all data underlying the findings have been made fully available, in compliance with the PLOS Data Policy.

5. Is the manuscript presented in an intelligible fashion and written in standard English?

Reviewer #3: Yes

Reviewer #5: Yes

Reviewer #6: Yes

Response: Reviewers #3, #5, and #6 thank you for your positive remarks regarding the clarity and quality of the manuscript's language. We are grateful for your recognition that the manuscript is written in Standard English and presented in an intelligible manner.

Reviewer #7: No

Response: Thank you for raising concerns regarding the manuscript's language. We recognize the importance of presenting the work clearly and ensuring it adheres to the highest linguistic standards. To address your concerns, we have thoroughly reviewed the manuscript to correct any typographical or grammatical errors and made necessary revisions. The language has been refined for better clarity, coherence, and precision to ensure it meets the required standard.

6. Review Comments to the Author

Reviewer #3: Dear Author,

I have previously evaluated the article titled "Indications and Endoscopic findings of upper gastrointestinal diseases in Africa: A systematic Review & Meta-analysis" with the number PONE-D-24-29678R1 and stated that its publication in your journal would contribute positively to the medical literature. I have also reviewed the criticisms of other referees and the authors' responses to the criticisms regarding the article sent to me for re-evaluation. I believe that all the corrections made by the authors after the criticisms of other referees are sufficient. In my opinion, the article would be appropriate to publish.

Best regards.

Response: Thank you very much for your kind and thoughtful evaluation of our manuscript. We truly appreciate your ongoing support and recognition of the contributions our article can make to the medical literature. We are grateful for your review of the revisions we made in response to the criticisms from other reviewers. It is reassuring to hear that you find the changes sufficient and that you consider the article appropriate for publication. Your positive assessment is greatly appreciated, and we remain committed to ensuring the quality and clarity of our work.

Reviewer #5: The authors have responded adequately to the review comments, addressing most of the concerns thoroughly. While there remains an unresolved point regarding the lack of specific data on the overall number of upper GI endoscopy cases in Africa and the study's proportionate representation within that total (Comment #6), the authors have explained that a comprehensive data source for all UGIE cases across Africa is unavailable. Given this limitation, the authors have taken steps to mitigate representational bias by including a geographically diverse range of studies and acknowledging this limitation in the revised manuscript.

The remaining unresolved point is noted as a future consideration and does not detract from the overall adequacy of the authors' responses. The reviewers find the authors' explanations and the additional transparency measures, including the full search strategy and supporting data files, sufficient to support the credibility and generalizability of the study's findings.

Response: Thank you for your valuable comments and recognition of our effort and for bringing attention to the concern, regarding the lack of specific data on the total number of upper gastrointestinal endoscopy (UGIE) cases in Africa. We would like to clarify that, of the total sample size of 120460 UGIE cases, we identified above 78% of those cases had at least one pathological findings in the included studies. From this subset, we identified 31 common endoscopic findings, including gastritis, esophageal cancer, peptic ulcer disease (PUD), and others. This information has been clearly presented in the Results section (Table 3 and 4), as well as discussed in discussion section. We have taken steps to reduce representational bias by including studies from a geographically diverse range of regions across Africa, despite the absence of a comprehensive data source for all UGIE cases. We acknowledge this limitation in the manuscript and have highlighted it for future consideration. We hope this clarification provides a comprehensive response to your concern and appreciate your recognition of our efforts to ensure the credibility and generalizability of our findings.

Reviewer #6: I have been included as a reviewer after some comprehensive recommendations were made by others and read the latest corrected text.

Response: Thank you for your valuable time and for reviewing the revised manuscript. We appreciate your consideration of the corrected text, and we are grateful for the comprehensive recommendations made by previous reviewers. Your feedback, along with the contributions of others, has been instrumental in enhancing the quality and clarity of our work

This systematic review and meta-analysis, while comprehensive, does have limitations that may affect the generalizability of the findings. 1) The reliance on institutional samples could lead to an overestimation of the prevalence of UGIE indications and findings, as these samples often consist of patients presenting with severe, persistent symptoms or advanced age. 2) The potential for publication bias should also be noted; studies reporting significant or positive results are more likely to be published, which may skew the pooled prevalence rates reported in the analysis. 3) Additionally, the restriction to English-language publications may have resulted in the exclusion of relevant data from non-English studies, further affecting the comprehensiveness of the findings.

Response: Thank you for your thoughtful comments. We appreciate your recognition of the comprehensiveness of our systematic review and meta-analysis. We also acknowledge the important limitations you have raised, which are relevant to the generalizability of the findings.

Reliance on Institutional Samples: We agree that the use of institutional samples may introduce bias, particularly by over representing cases of upper gastrointestinal disease. We have recognized this limitation in the manuscript and have made it clear that the findings may not fully reflect the broader population of upper gastrointestinal disease patients, especially those with rural, or from primary care settings.

Publication Bias: We also acknowledge the potential for publication bias, as studies with significant or positive results are more likely to be published. This limitation is noted in the manuscript, and we emphasize that the pooled prevalence rates may be influenced by this bias, affecting the overall validity of our estimates.

Restriction to English-Language Publications: We recognize that restricting the review to English-language publications may exclude relevant data from non-English studies. This is indeed a limitation of the review process, and we have transparently stated this in the manuscript to highlight the potential impact on the comprehensiveness of our findings.

4)Moreover, Africa is a huge continent where life style, socioeconomics or eating habits varies from north to south + east to west. Instead of generalisation of the whole continent data, it would be better if authors would be able to focus on regional findings. However, the solution of this problem will bring some workload as further dataset analysis that the authors may not prefer.

Response: Thank you for your insightful comment regarding the diversity within Africa and the suggestion to focus on regional findings. We agree that Africa is a vast continent with significant variations in lifestyle, socioeconomics, and dietary habits from region to region. In response to your suggestion, we have made an effort to analyze regional differences in the data where possible and have included regional distribution in our result by adding new table 4, and made discussion about the regional distribution variation.

5) The discussion addresses the frequency of all symptoms and outcomes and compares them as numbers. However, does not discuss substantially the possible causes that those differences arise.

Response: Thank you for your comment regarding the discussion of the frequencies of symptoms and outcomes. We appreciate your suggestion to delve deeper into the possible causes behind the observed differences. In response, we have expanded the discussion to include potential factors that may contribute to the variations, such as regional differences in healthcare access, socio-economic factors, lifestyle, dietary habits, and other underlying medical conditions that could influence the prevalence and presentation of symptoms.

Reviewer #7: Reviewer Comments for: Indications and Endoscopic Findings of Upper Gastrointestinal Diseases in Africa – A systematic Review and Meta-analysis - PONE-D-24-29678R1

1. The abstract should be re-written based on changes suggested below.

Response: Thank you for your suggestion regarding the abstract.

---

## [Editor Report · Decision Letter 2]

11 Feb 2025

Indications and Endoscopic findings of upper gastrointestinal diseases in Africa: A systematic Review & Meta-analysis

PONE-D-24-29678R2

Dear Dr. Seid,

We’re pleased to inform you that your manuscript has been judged scientifically suitable for publication and will be formally accepted for publication once it meets all outstanding technical requirements.

Kind regards,

Muhammad Salman Bashir, M.S.C

Academic Editor

PLOS ONE
---

## [Editor Report · Acceptance letter]

PONE-D-24-29678R2

PLOS ONE

Dear Dr. Abdu,

I'm pleased to inform you that your manuscript has been deemed suitable for publication in PLOS ONE. Congratulations! Your manuscript is now being handed over to our production team.

Kind regards,

on behalf of

Dr. Muhammad Salman Bashir

Academic Editor

PLOS ONE
